

# The complete mitochondrial genome of the grooved carpet shell, *Ruditapes decussatus* (Bivalvia, Veneridae)

Fabrizio Ghiselli[1],[*], Liliana Milani[1],[*], Mariangela Iannello[1], Emanuele Procopio[1], Peter L. Chang[2], Sergey V. Nuzhdin[2] and Marco Passamonti[1]

[1] Department of Biological, Geological and Environmental Sciences, University of Bologna, Italy, Bologna, Italy
[2] Department of Biological Sciences, Program in Molecular and Computational Biology, University of Southern California, Los Angeles, CA, USA
[*] These authors contributed equally to this work.

Corresponding author
Fabrizio Ghiselli,
fabrizio.ghiselli@unibo.it

## ABSTRACT

Despite the large number of animal complete mitochondrial genomes currently available in public databases, knowledge about mitochondrial genomics in invertebrates is uneven. This paper reports, for the first time, the complete mitochondrial genome of the grooved carpet shell, *Ruditapes decussatus*, also known as the European clam. *Ruditapes decussatus* is morphologically and ecologically similar to the Manila clam *Ruditapes philippinarum*, which has been recently introduced for aquaculture in the very same habitats of *Ruditapes decussatus*, and that is replacing the native species. Currently the production of the European clam is almost insignificant, nonetheless it is considered a high value product, and therefore it is an economically important species, especially in Portugal, Spain and Italy. In this work we: (i) assembled *Ruditapes decussatus* mitochondrial genome from RNA-Seq data, and validated it by Sanger sequencing; (ii) analyzed and characterized the *Ruditapes decussatus* mitochondrial genome, comparing its features with those of other venerid bivalves; (iii) assessed mitochondrial sequence polymorphism (SP) and copy number variation (CNV) of tandem repeats across 26 samples. Despite using high-throughput approaches we did not find evidence for the presence of two sex-linked mitochondrial genomes, typical of the doubly uniparental inheritance of mitochondria, a phenomenon known in ~100 bivalve species. According to our analyses, *Ruditapes decussatus* is more genetically similar to species of the Genus Paphia than to the congeneric *Ruditapes philippinarum*, a finding that bolsters the already-proposed need of a taxonomic revision. We also found a quite low genetic variability across the examined samples, with few SPs and little variability of the sequences flanking the control region (Largest Unassigned Regions (LURs). Strikingly, although we found low nucleotide variability along the entire mitochondrial genome, we observed high levels of length polymorphism in the LUR due to CNV of tandem repeats, and even a LUR length heteroplasmy in two samples. It is not clear if the lack of genetic variability in the mitochondrial genome of *Ruditapes decussatus* is a cause or an effect of the ongoing replacement of *Ruditapes decussatus* with the invasive *Ruditapes philippinarum*, and more analyses, especially on nuclear sequences, are required to assess this point.

## INTRODUCTION

Despite a large number of animal complete mitochondrial genomes (mtDNAs) being available in public databases (>55,000 in GenBank), up to now sequencing has been focused mostly on vertebrates (~50,000 in GenBank), and the current knowledge about mitochondrial genomics in invertebrates—with the notable exception of few model organisms (e.g., Drosophila and *Caenorhabditis elegans*)—is uneven. To better understand invertebrate mitochondrial biology—and, most importantly, mitochondrial biology and evolution in general—it is necessary to adopt a more widespread approach in gathering and analyzing data. Failing to do so would bias our knowledge toward a few taxonomic groups, with the risk of losing a big part of the molecular and functional diversity of mitochondria. Actually, despite maintaining its core features in terms of genetic content, mtDNA in Metazoa shows a wide range of variation in some other traits such as, for example, genome architecture, abundance of unassigned regions (URs)—namely regions with no assigned product (protein, RNA)—repeat content, gene duplications, introns, UTRs, and even additional coding genes (see *Breton et al., 2014* for a review) or genetic elements (e.g., small RNAs, see *Pozzi et al., 2017*). All this emerging diversity is in sharp contrast with the—at this point outdated—textbook notion about mtDNAs role being limited to the production of a few subunits of the protein complexes involved in oxidative phosphorylation.

This paper reports, for the first time, the complete mitochondrial genome of the grooved carpet shell, *Ruditapes decussatus* (Linnaeus, 1758). *Ruditapes decussatus*—also known as the European clam—is distributed all over the Mediterranean coasts, as well as on the Atlantic shores, from Lofoten Islands (Norway) to Mauritania, including the British Isles. *Ruditapes decussatus* lives in warm coastal waters, especially in lagoons, and it is morphologically and ecologically similar to the Manila clam *Ruditapes philippinarum*, which has been recently introduced for aquaculture in the very same habitats of *Ruditapes decussatus*. *Ruditapes philippinarum*, native from the Philippines, Korea, and Japan, was accidentally introduced into North America in the 1930s, and from there it was purposely introduced in France (1972), UK (1980), and Ireland (1982) for aquaculture purposes (*Gosling, 2003*). According to historical records, *Ruditapes decussatus* was one of the most important species for aquaculture in Europe, but overfishing, irregular yields, recruitment failure, and outbreaks of bacterial infection pushed the producers to introduce *Ruditapes philippinarum*; Italy imported large quantities of *Ruditapes philippinarum* seed from UK in 1983 and 1984. Compared to the European clam, the Manila clam turned out to be faster growing, more resistant to disease, to have a more extended breeding period and a greater number of spawning events, and to begin sexual maturation earlier (i.e., at a smaller size). Upon introduction of the more robust *Ruditapes philippinarum*, *Ruditapes decussatus* suffered a population decline in the Southwestern Europe (*Arias-Pérez et al., 2016*), and

currently the production of the European clam is almost insignificant. Nonetheless the grooved carpet shell is considered a high value product, and therefore it is an economically important species, especially in Portugal, Spain and Italy (*Gosling, 2003*; *Leite et al., 2013*; *de Sousa et al., 2014*).

Molluscs in general, and bivalves in particular, exhibit an extraordinary degree of mtDNA variability and unusual features, such as: large mitochondrial genomes (up to ~47Kb), high proportion of URs (i.e., number of base pairs annotated as URs over the total mtDNA length), novel protein coding genes with unknown function, frequent and extensive gene rearrangement, and differences in strand usage (*Gissi, Iannelli & Pesole, 2008*; *Breton et al., 2011*; *Ghiselli et al., 2013*; *Milani et al., 2014b*; *Plazzi, Puccio & Passamonti, 2016*). Moreover, mitochondrial genome size varies among bivalves because of gene duplications and losses (*Serb & Lydeard, 2003*; *Passamonti et al., 2011*; *Ghiselli et al., 2013*), and sometimes genes are fragmented as in the case of ribosomal genes in oysters (*Milbury et al., 2010*). The most notable feature of bivalve mtDNA is the doubly uniparental inheritance (DUI) system of transmission (*Skibinski, Gallagher & Beynon, 1994a, 1994b*; *Zouros et al., 1994a, 1994b*). Under DUI, two different mitochondrial lineages (and their respective genomes) are transmitted to the progeny: one is inherited from the egg (female-transmitted or F-type mtDNA), the other is inherited from the spermatozoon (male-transmitted or M-type mtDNA). Following fertilization, the early embryo is heteroplasmic, but the type of mitochondria present in the adult is tightly linked to its sex. Females are commonly homoplasmic for F, while males are heteroplasmic with the following distribution of mtDNA types: the germ line is homoplasmic for the M-type (which will be transmitted via sperm to male progeny), the soma is heteroplasmic to various degrees, depending on tissue type and/or species (*Ghiselli, Milani & Passamonti, 2011*; *Zouros, 2013*). To date, the only known animals exhibiting DUI are about 100 species of bivalve molluscs (*Gusman et al., 2016*). This natural and evolutionarily stable heteroplasmic system can be extremely useful to investigate several aspects of mitochondrial biology (see *Passamonti & Ghiselli, 2009*; *Breton et al., 2014*; *Milani & Ghiselli, 2015*; *Milani, Ghiselli & Passamonti, 2016*). Indeed, despite the fact that many aspects of DUI are still unknown, there is evidence that DUI evolved from a strictly maternal inheritance (SMI) system (*Milani & Ghiselli, 2015*; *Milani, Ghiselli & Passamonti, 2016*), by modifications of the molecular machinery involved in mitochondrial inheritance, through as-yet-unknown specific factors (see *Diz, Dudley & Skibinski, 2012*; *Zouros, 2013* for proposed models). The detection of DUI is not a straightforward process, especially using PCR-based approaches: given that the divergence between F and M genomes is often comparable to the distance between mtDNAs of different classes of Vertebrates, primers may fail to amplify one of the two mtDNAs, yielding a false-negative result. Moreover, M-type mtDNA can be rare in somatic tissues, so it may be difficult to amplify from animals sampled outside of the reproductive season, when gonads are absent (thoroughly discussed in *Theologidis et al., 2008*). High-throughput sequencing (HTS) approaches can overcome such problems, because a prior knowledge of the mtDNA sequence is not needed, and low-copy variants can be easily unveiled (see *Ju et al., 2011*; *King et al., 2014*). Until now, HTS has been scarcely

utilized to study mitochondrial transcriptomes and genomes (*Pesole et al., 2012*; *Smith, 2013*), even if it showed very good potential (*Lubośny et al., 2017/2*; *Yuan et al., 2016*). In this work we: (i) assembled *Ruditapes decussatus* mitochondrial genome from RNA-Seq data, and validated it by Sanger sequencing, (ii) analyzed and characterized *Ruditapes decussatus* mitochondrial genome, comparing its features with those of other venerid bivalves; (iii) assessed mitochondrial sequence polymorphism (SP) and structural variants—copy number variation (CNV) of tandem repeats—among the sampled animals.

## MATERIALS AND METHODS

### Sampling

The 26 *Ruditapes decussatus* specimens used in this study were collected from the Northern Adriatic Sea, in the river Po Delta region (Sacca di Goro, approximate GPS coordinates: 44°50′06″N, 12°17′55″E) at the end of July 2011, during the spawning season. Each individual was dissected, and gonadal liquid collected with a glass capillary tube. All the samples showed ripe gonads, consistently with the time of the year when the sampling occurred. The gonadal liquid was checked under a light microscope to assess the sex of the individual, and to make sure that the sample consisted of mature gametes. Both the gamete samples and the clam bodies were flash-frozen in liquid nitrogen, and stored at −80 °C, until nucleic acid extraction. Table S1 shows the sample list, and details about data availability.

### RNA-Seq

In total, 12 samples (six males and six females) were used for RNA-Seq. Total RNA extraction and library preparation were performed following the protocol described in *Mortazavi et al. (2008)*, with the modifications specified in *Ghiselli et al. (2012)*. The 12 samples were indexed, pooled and sequenced in two lanes (two technical replicates) of Illumina GA IIx, using 76 bp paired-end reads.

### De novo assembly

The mitochondrial genome of *Ruditapes decussatus* was not available in the databases, so we used the transcriptome data to generate a draft to be used as a guide for Sanger sequencing. Illumina reads from all 12 samples were pooled and compared to a set of 20 Bivalvia mitochondrial genomes to identify reads with mitochondrial origin. Alignment was done using BLASTN. All reads with similarity yielding *E*-value < 1E-5 were then assembled into contigs using the A5 pipeline (version 2013032; *Tritt et al., 2012*) and joined into scaffolds using CAP3 (*Huang & Madan, 1999*). For the quality check step, we applied a PHRED Q-score cutoff threshold of 33; the other A5 parameters were set as default. CAP3 was run with default settings as well.

### Sanger validation

In total, 14 *Ruditapes decussatus* samples from the same collection campaign—sexed, and stored at −80 °C—were used for DNA extraction. DNA from the gonadic tissue was

extracted using the Qiagen DNeasy kit. Primers for mtDNA amplification were designed based on contigs obtained from RNA-Seq matching venerid mtDNA sequences, then the "primer walking" method was used to Sanger-sequence the complete mitochondrial genome of *Ruditapes decussatus*. The primers were designed with the software Primer3 (*Rozen & Skaletsky, 2000*) and tested on several samples, then a female was chosen as reference sample for Sanger validation of mtDNA *de novo* assembly. In addition, we amplified the largest unassigned region (LUR) of 13 females to assess its variability (see Results and Discussion). The list of the primers and their sequences are reported in Table S2. PCR reactions were performed in a final volume of 50 µl using the GoTaq Flexi DNA Polymerase Kit (Promega, Madison, WI, USA), on a 2720 Thermal Cycler (Applied Biosystem, Foster City, CA, USA). The PCR reactions were set as follows: initial denaturation 95 °C for 1 min, then 30 cycles of amplification (denaturation 95 °C for 1 min, annealing 48–60 °C for 1 min, extension 72 °C for 1 min/kb), then the final extension at 72 °C for 5 min. PCR products were checked by electrophoretic run on 1% agarose gel, and then purified using the DNA Clean & Concentrator-25 kit (Zymo Research, Irvine, CA, USA).

Sanger sequencing was performed by Macrogen Inc. (http://www.macrogen.com).

Sequences were aligned with the software MEGA 6.0 (*Tamura et al., 2013*), using the contigs obtained by RNA-seq as a reference.

## Annotation

Open reading frames (ORFs) were identified with ORF finder (*Wheeler et al., 2005*). Alternative start codons were considered functional because they are common in Bivalvia. ORFs were annotated starting from the first available start codon (ATG, ATA, or ATC) downstream of the preceding gene, and ending with the first stop codon in frame (TAA or TAG). tRNA genes and their structure were identified with MITOS (*Bernt et al., 2013*) and ARWEN (*Laslett & Canback, 2008*). Secondary structures were predicted using the RNAFold Server, included in the ViennaRNA Web Services (http://rna.tbi.univie.ac.at/; *Gruber et al., 2008*); the folding temperature was set at 16 °C which is the average annual temperature of the water from which the *Ruditapes decussatus* specimens used in this work were fished (download RNAFold results from figshare: https://ndownloader.figshare.com/files/8387672). tRNAs and other secondary structures were drawn with the software Varna GUI (*Darty, Denise & Ponty, 2009*). Ribosomal small subunit (*rrnS*) and large subunit (*rrnL*) were identified with BLASTN, and annotated considering the start and the end of the adjacent genes as the boundaries of the rRNA genes. Non-genic regions were annotated as URs. In order to identify the putative D-loop/control region (CR), we analyzed the LUR with the MEME suite (*Bailey et al., 2009*) to find DNA motifs using the following bivalve species as comparison: *Acanthocardia tuberculata, Arctica islandica, Coelomactra antiquata, Fulvia mutica, Hiatella arctica, Loripes lacteus, Lucinella divaricata, Lutraria rhynchaena, Meretrix lamarckii* (F-type), *Meretrix lamarckii* (M-type), *Meretrix lusoria, Meretrix lyrata, Meretrix meretrix, Meretrix petechialis, Moerella iridescens, Nuttallia olivacea, Paphia amabilis, Paphia euglypta, Paphia textile, Paphia undulata, Ruditapes philippinarum* (F-type), *Ruditapes philippinarum* (M-type),

*Semele scabra, Sinonovacula constricta, Solecurtus divaricatus, Solen grandis, Solen strictus, Soletellina diphos* and the sea urchin *Strongylocentrotus purpuratus* (Echinoidea, Strongylocentrotidae). The list of the species used in the phylogenetic analysis and in the comparative analyses of DNA motifs, sequence similarity, and gene order are available in Table S3. The GOMo (Gene Ontology for Motifs; *Buske et al., 2010*) tool of the MEME suite was used to assign GO terms to the motifs discovered.

The number of repeats in the LUR of the reference sample (F4) was calculated with tandem repeat finder (http://tandem.bu.edu/trf/trf.html), since the complete LUR sequence was available (download tandem repeat finder results from Figshare: https://ndownloader.figshare.com/files/8387666). In the other cases, in which the LUR could not be sequenced without gaps, the number of repeats was inferred from agarose gel electrophoresis.

### Other analyses

Comparisons among venerid complete mtDNAs were performed with BLAST Ring Image Generator (BRIG) (*Alikhan et al., 2011*) and Easyfig (*Sullivan, Petty & Beatson, 2011*). Descriptive statistics were obtained with MEGA v6.0 (*Tamura et al., 2013*), except for the codon usage table, which was obtained with the Sequence Manipulation Suite (*Stothard, 2000*). SP assessment from RNA-Seq reads was performed with the Genome Analysis Toolkit (GATK, *McKenna et al., 2010*), with the Sanger-sequenced mtDNA as reference. For SP discovery and genotyping we used standard hard filtering parameters or variant quality score recalibration (*DePristo et al., 2011*). The MitoPhast pipeline (*Tan et al., 2015*) was used to obtain the Maximum Likelihood (ML) tree, which was visualized with Evolview v2 (*He et al., 2016*). Briefly, MitoPhast takes as input GenBank files (.gb), extracts the coding sequences, profiles the sequences with Pfam (*Finn et al., 2016*) and PRINTS (*Attwood et al., 2003*), performs a multiple sequence alignment with Clustal Omega (*Sievers et al., 2011*), removes poorly aligned regions with trimAl (*Capella-Gutiérrez, Silla-Martinez & Gabaldón, 2009*), concatenates the coding sequences, performs data partitioning and model selection, and then carries out a ML analysis using RAxML (*Stamatakis, 2014*). The species used in the ML analysis, and their GenBank Accession Numbers are listed in Table S3. Amino acid sequences of three different *cox3* ORFs inferred from Sanger sequencing and GATK polymorphism data were analyzed with InterProScan (*Jones et al., 2014*).

## RESULTS

### De novo assembly and Sanger validation

Despite using HTS on extracts of ripe gonads (i.e., mature gametes), and multiple assembly strategies (see Discussion for details) we could not find evidence for DUI. The de novo assembly process produced 9 contigs, of which 8 included multiple genes, and one included a single gene (see Table 1). The sequences of the contigs in FASTA format are available on figshare (https://ndownloader.figshare.com/files/8906839). In four cases (Contigs 1, 3, 6, and 7) a clear polyadenylation signal was present, in other four cases (Contigs 2, 5, 8, and 9) it was not. Contig 4, the only one including a single gene (*cox3*), ends with just 8 As, so it is not clear if a polyadenylation signal is present in this case.

**Table 1 Features of the contigs obtained by *de novo* assembly of mtDNA.**

| Contig | Length | Gene content | Poly-A | Notes |
|---|---|---|---|---|
| 1 | 6,794 | atp6_nd3_nd5_cox1_ tRNA-Leu1_nd1_nd2_nd4L | Yes | Chimeric assembly. The contiguity between *nd5* and *cox1* is an artifact |
| 2 | 1,884 | rrnS_cox3 | No | – |
| 3 | 1,288 | atp6_nd3 | Yes | – |
| 4 | 1,663 | cox3 | ? | The contig ends with just 8 As |
| 5 | 1,934 | atp8_nd4_tRNA-His_tRNA-Glu_ tRNA-Ser2_tRNA-Tyr | No | – |
| 6 | 1,831 | atp8_nd4_tRNA-His | Yes | – |
| 7 | 5,478 | cox2_tRNA-Ile_nd4L_nd2_nd1_ tRNA-Leu1_cox1 | Yes | There is a polyadenylation signal (56 As) after the *cox2* gene |
| 8 | 2,879 | cytb_rrnL | No | – |
| 9 | 952 | nd6_tRNA-Lys_tRNA-Val_tRNA-Phe_ tRNA-Trp_tRNA-Arg_tRNA-Leu2 | No | – |

In Contig 7 (that includes *cox2, tRNA-Ile, nd4L, nd2, nd1, tRNA-Leu1*, and *cox1* genes) there is a polyadenylation signal (56 As) after the *cox2* gene.

The nine contigs were used as a scaffold for the primer walking procedure used for Sanger validation of the *de novo* assembly. We first tried to connect the contigs designing primers close to the 5′ and 3′ ends of each contig and pairing them following the gene order of Paphia, because the sequence of genes in the contigs suggested that *Ruditapes decussatus* gene order might have been similar. During such process, Contig 1 turned out to be a chimeric assembly between two non-contiguous portions of the mtDNA, one including *atp6, nd3,* and *nd5*, the other including *cox1, tRNA-Leu1, nd1, nd2,* and *nd4L*. Once we amplified and sequenced the portions of mtDNA between the contigs, we proceeded with the Sanger resequencing of the remaining parts.

## Annotation and mtDNA features

The mitochondrial genome contains 13 protein-coding genes, and in the reference female is 18,995 bp long (Fig. 1); the gene arrangement and other details are shown in Table 2. All genes are located on the heavy strand, and in addition to the classic start codon ATG (Met), the alternative start codons ATA (Met) and ATC (Ile) are present. The most frequently used start codons are: ATA (*cox1, nd1, nd4L, cox2, cob, atp8, nd4*), and ATG (*nd2, atp6, nd3, nd5, nd6, cox3*). The stop codons found are TAG (*cox1, nd2, nd4L, cox2, cytb, nd4*) and TAA (*nd1, atp6, nd3, atp8, nd6*). The *nd4* gene has an incomplete stop codon (TA-). 22 tRNA genes were identified, including two tRNAs for leucine, tRNA-Leu1 (TAG) and tRNA-Leu2(TAA), and two for serine, tRNA-Ser1(TCT) and tRNA-Ser2 (TGA), both showing degenerate D-arm branches. tRNA structures are shown in Fig. S1. The two rRNAs, *rrnS* and *rrnL*, were both identified: the *rrnS* is located between *cox3* and *cox1*, while *rrnL* is between *cytb* and *atp6*. URs were identified on the basis of unannotated spaces between different genes; we found 24 URs (Table 3).

The analysis of the nucleotide composition points out that the mitochondrial genome of this bivalve species exhibits high A+T content, totaling 63% vs 37% G+C. The

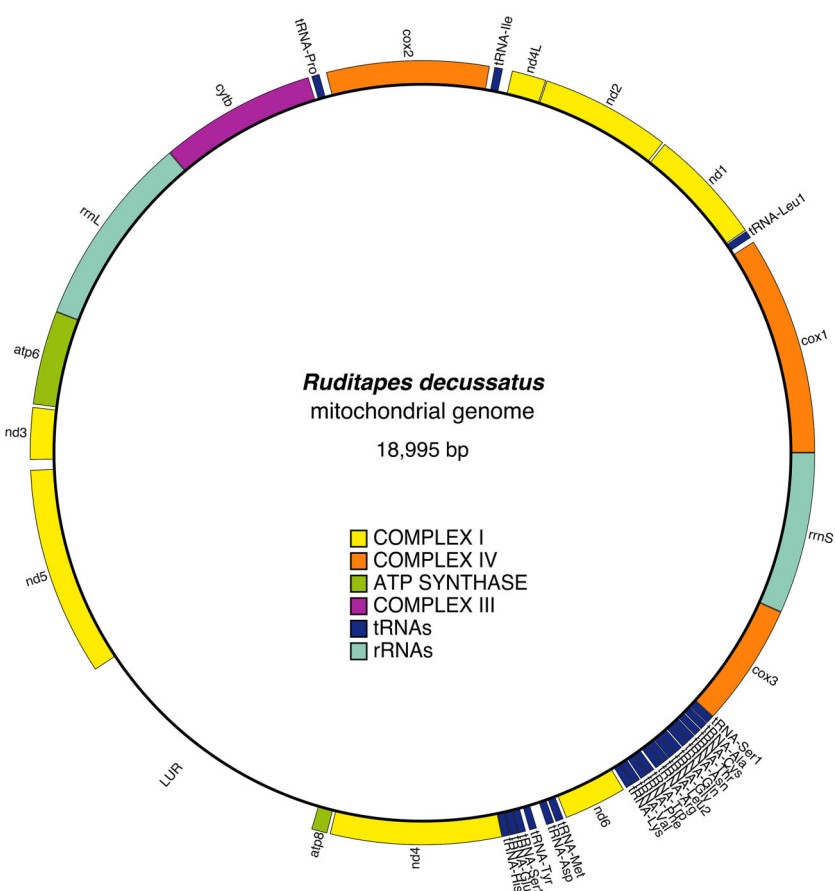

**Figure 1** *Ruditapes decussatus* **mtDNA gene arrangement.**

minimum values of A+T are found in *cytb* (60.1%) and *nd4* (61%). The nucleotide composition of every gene is shown in Table 4. According to the analysis above, both A and T occur very frequently at the third position of codons (64.6% on average of A+T), while the less frequent base in third position is C (12%). The most used codons are UUU (Phe), counted 269 times, and UUA (Leu) counted 210 times (6.78% and 5.29% of the total, respectively), while the less used codons are CGC (Arg) counted 6 times (0.15%), ACC (Thr) and CCG (Pro) each counted 16 times (0.4%) (Table 5). Only in four cases over 20 (Lys, Leu, Gln, Val), the most frequently used codon matches the correspondent mitochondrial tRNA anticodon.

The UR11 is the LUR and is located between *atp8* and *nd5* (Figs. 1 and 2A). The LUR of the female used for whole mtDNA Sanger sequencing (i.e., the reference female, F4) is 2,110 bp long, and includes 6.5 repeated sequences—each repeat having a length of 54 bp —localized in the 3′ region of the LUR, just upstream the *atp8* gene (Fig. 2A). DNA secondary structure analysis predicted three stem-loop structures in such region (Fig. 2B and Supplemental Information files on figshare: https://ndownloader.figshare.com/files/ 8387672), with a change in Gibbs free energy (ΔG) of −71.38 Kcal/mol. We also amplified and sequenced the LUR of 13 more females. We were not able to completely sequence LURs longer than 2,110 bp, because of the known difficulties in Sanger sequencing of

**Table 2 MtDNA gene arrangement of *Ruditapes decussatus*.**

| Name | Type | Start | Stop | Length (bp) | Start | Stop | Anticodon |
|------|------|-------|------|-------------|-------|------|-----------|
| cox1 | Coding | 1 | 1,716 | 1,716 | ATA | TAG | |
| tRNA-Leu1 | tRNA | 1,754 | 1,815 | 62 | | | TAG |
| nd1 | Coding | 1,822 | 2,739 | 918 | ATA | TAA | |
| nd2 | Coding | 2,755 | 3,774 | 1,020 | ATG | TAG | |
| nd4l | Coding | 3,780 | 4,052 | 273 | ATA | TAG | |
| tRNA-Ile | tRNA | 4,125 | 4,190 | 66 | | | GAT |
| cox2 | Coding | 4,228 | 5,499 | 1,272 | ATA | TAG | |
| tRNA-Pro | tRNA | 5,553 | 5,616 | 64 | | | TGG |
| cytb | Coding | 5,641 | 6,864 | 1,224 | ATA | TAG | |
| rrnL | rRNA | 6,865 | 8,385 | 1,521 | | | |
| atp6 | Coding | 8,386 | 9,123 | 738 | ATG | TAA | |
| nd3 | Coding | 9,145 | 9,552 | 408 | ATG | TAA | |
| nd5 | Coding | 9,631 | 11,268 | 1,638 | ATG | TAG | |
| atp8 | Coding | 13,379 | 13,504 | 126 | ATA | TAA | |
| nd4 | Coding | 13,526 | 14,865 | 1,340 | ATA | TA- | |
| tRNA-His | tRNA | 14,866 | 14,928 | 63 | | | GTG |
| tRNA-Glu | tRNA | 14,929 | 14,990 | 62 | | | TTC |
| tRNA-Ser2 | tRNA | 14,991 | 15,052 | 62 | | | TGA |
| tRNA-Tyr | tRNA | 15,081 | 15,140 | 60 | | | GTA |
| tRNA-Asp | tRNA | 15,218 | 15,280 | 63 | | | GTC |
| tRNA-Met | tRNA | 15,294 | 15,358 | 65 | | | CAT |
| nd6 | Coding | 15,380 | 15,874 | 495 | ATG | TAA | |
| tRNA-Lys | tRNA | 15,897 | 15,959 | 63 | | | TTT |
| tRNA-Val | tRNA | 15,960 | 16,021 | 62 | | | TAC |
| tRNA-Phe | tRNA | 16,030 | 16,092 | 63 | | | GAA |
| tRNA-Trp | tRNA | 16,093 | 16,155 | 63 | | | TCA |
| tRNA-Arg | tRNA | 16,171 | 16,232 | 62 | | | TCG |
| tRNA-Leu2 | tRNA | 16,233 | 16,295 | 63 | | | TAA |
| tRNA-Gly | tRNA | 16,297 | 16,358 | 62 | | | TCC |
| tRNA-Gln | tRNA | 16,359 | 16,427 | 69 | | | TTG |
| tRNA-Asn | tRNA | 16,435 | 16,497 | 63 | | | GTT |
| tRNA-Thr | tRNA | 16,498 | 16,560 | 63 | | | TGT |
| tRNA-Cys | tRNA | 16,565 | 16,626 | 62 | | | GCA |
| tRNA-Ala | tRNA | 16,632 | 16,696 | 65 | | | TGC |
| tRNA-Ser1 | tRNA | 16,698 | 16,764 | 67 | | | TCT |
| cox3 | Coding | 16,765 | 17,730 | 966 | ATG | TAA | |
| rrnS | rRNA | 17,731 | 18,995 | 1,265 | | | |

**Note:**
The anticodon of tRNAs are reported in the 5'-3' direction.

**Table 3 Unassigned regions (URs).**

| UR name | Start | Stop | Length (bp) |
|---|---|---|---|
| UR1 | 1,717 | 1,753 | 37 |
| UR2 | 1,816 | 1,821 | 6 |
| UR3 | 2,740 | 2,754 | 15 |
| UR4 | 3,775 | 3,779 | 5 |
| UR5 | 4,053 | 4,124 | 72 |
| UR6 | 4,191 | 4,227 | 37 |
| UR7 | 5,500 | 5,552 | 53 |
| UR8 | 5,617 | 5,640 | 24 |
| UR9 | 9,124 | 9,144 | 21 |
| UR10 | 9,553 | 9,630 | 78 |
| UR11 (LUR) | 11,269 | 13,378 | 2,110 |
| UR12 | 13,505 | 13,525 | 21 |
| UR13 | 15,053 | 15,080 | 28 |
| UR14 | 15,141 | 15,217 | 77 |
| UR15 | 15,281 | 15,293 | 13 |
| UR16 | 15,359 | 15,379 | 21 |
| UR17 | 15,875 | 15,896 | 22 |
| UR18 | 16,022 | 16,029 | 8 |
| UR19 | 16,156 | 16,170 | 15 |
| UR20 | 16,296 | 16,296 | 1 |
| UR21 | 16,428 | 16,434 | 7 |
| UR22 | 16,561 | 16,564 | 4 |
| UR23 | 16,627 | 16,631 | 5 |
| UR24 | 16,697 | 16,697 | 1 |

regions including multiple repeats. The sequence alignment of the 13 LURs is available for download from figshare (https://ndownloader.figshare.com/files/8360789). LUR lengths, inferred from gel electrophoresis, are reported in Table 6, and they range from 2,000 to 5,000 bp. Two females (F3 and F17) showed length heteroplasmy of the LUR. The portion of the genome occupied by URs varies between 14.11% and 29.38%, depending on LUR length. The analysis with MEME (output shown in Figs. S2 and S3) unveiled two motifs (Fig. 2C) that show a strong conservation within the Veneridae family, and with *S. purpuratus*. The sea urchin was included in the analysis because *Cao et al. (2004)* reported a match between some motifs found in the CR of the marine mussels *Mytilus edulis* and *Mytilus galloprovincialis* with regulatory elements of the sea urchin CR. Accordingly, the search with GOMo assigned a series of GO terms related to transcription to the two motifs (Table S4).

## Polymorphism

Table 7 (top) shows the statistics associated with the SP analysis performed with GATK on the 12 samples used for RNA-Seq, with the Sanger-sequenced mtDNA as reference. Overall, 257 SPs were called, of which 145 (56.4%) were located in coding sequences

**Table 4 Nucleotide composition.**

| Name | Length (bp) | T (%) | C (%) | A (%) | G (%) | A+T (%) | T3 (%) | C3 (%) | A3 (%) | G3 (%) | A3+T3 (%) |
|------|-------------|-------|-------|-------|-------|---------|--------|--------|--------|--------|-----------|
| cox1 | 1,716 | 35.8 | 15.5 | 25.8 | 22.9 | 61.6 | 39 | 12.1 | 28.0 | 21.3 | 67.0 |
| nd1 | 918 | 38.7 | 12.5 | 24.0 | 24.8 | 62.7 | 38 | 10.1 | 30.7 | 21.2 | 68.7 |
| nd2 | 1,020 | 38.3 | 11.0 | 24.8 | 25.9 | 63.1 | 35 | 11.5 | 29.4 | 24.4 | 64.4 |
| nd4l | 273 | 39.9 | 12.8 | 25.3 | 22.0 | 65.2 | 34 | 14.3 | 30.8 | 20.9 | 64.8 |
| cox2 | 1,272 | 29.7 | 14.8 | 29.1 | 26.4 | 58.8 | 30 | 15.3 | 27.4 | 27.6 | 57.4 |
| cob | 1,224 | 37.4 | 17.2 | 22.7 | 22.6 | 60.1 | 41 | 14.7 | 21.8 | 22.1 | 62.8 |
| rrnL | 1,749 | 33.2 | 11.5 | 32.6 | 22.6 | 65.8 | 33 | 10.6 | 33.4 | 23.0 | 66.4 |
| atp6 | 510 | 42.0 | 15.7 | 20.8 | 21.6 | 62.8 | 45 | 13.5 | 21.8 | 20.0 | 66.8 |
| nd3 | 408 | 39.5 | 11.0 | 24.8 | 24.8 | 64.3 | 33 | 11.0 | 30.1 | 25.7 | 63.1 |
| nd5 | 1,638 | 37.6 | 11.7 | 27.7 | 23.0 | 65.3 | 35 | 11.0 | 34.2 | 19.8 | 69.2 |
| atp8 | 126 | 44.4 | 11.9 | 19.0 | 24.6 | 63.4 | 45 | 4.8 | 23.8 | 26.2 | 68.8 |
| nd4 | 1,340 | 38.9 | 12.9 | 22.1 | 26.1 | 61.0 | 41 | 10.8 | 24.9 | 23.5 | 65.9 |
| nd6 | 495 | 39.2 | 12.1 | 23.0 | 25.7 | 62.2 | 38 | 13.9 | 27.9 | 20.0 | 65.9 |
| cox3 | 966 | 36.9 | 12.7 | 24.8 | 25.6 | 61.7 | 39 | 9.6 | 28.6 | 23.0 | 67.6 |
| rrnS | 1,265 | 32.7 | 12.3 | 32.9 | 22.1 | 65.6 | 35 | 13.5 | 31.6 | 19.5 | 66.6 |
| All coding | 14,920 | 36.3 | 13.2 | 26.5 | 24.0 | 63.0 | 37 | 12.0 | 28.9 | 22.4 | 65.7 |
| All rRNAs | 3,014 | 32.9 | 23.8 | 32.7 | 22.3 | 65.7 | | | | | |
| All tRNAs | 1,394 | 35.4 | 12.8 | 30.2 | 21.7 | 65.6 | | | | | |
| All URs | 2,681 | 28.2 | 14.1 | 34.1 | 23.6 | 62.3 | | | | | |
| All genic DNA | 16,314 | 36.2 | 13.2 | 26.8 | 23.8 | 63.0 | | | | | |
| All DNA | 18,995 | 35.1 | 13.3 | 27.9 | 23.7 | 63.0 | | | | | |

**Note:**
URs, unassigned regions.

(CDS). Interestingly, most of the SPs were called because of private alleles of one single male specimen (mRDI01). More in detail, 151 SPs out of 257 (58.7%) along the whole mtDNA sequence, and 103 SPs out of 145 (71%) in CDS, were private of mRDI01. In CDS, if we exclude the SPs associated with this male, the number of polymorphisms drops to 42 over 14,920 bp of coding mtDNA (GATK output in VCF format and a detailed list of SPs in tabular format is available on figshare: https://ndownloader.figshare.com/files/8902537), of which 18 are represented by indels, 6 of which are located in 4 different coding genes: one each in cox1, cytb, and nd5, plus 3 in cox3 (see Table 8). A file showing the ORF generated by the different variants of cox3, and alignments between them is available on figshare (https://ndownloader.figshare.com/files/8402471). Table 7 (bottom) shows the number of SPs in males, in males except mRDI01, and in females both along the whole mtDNA, and in CDS. The number in brackets represent the number of private SPs for each category.

## Comparison with other veneridae

Figure 3 shows the *Ruditapes decussatus* mtDNA map (external gray circle), and the BLASTN identity (colored inner circles) with complete mtDNAs of other 10 venerid species (see list in Table S3). Figure 4 shows the ML tree obtained with the MitoPhast

**Table 5 Codon usage.**

| Amino acid | Codon | # | Frequency | %TOT | Amino acid | Codon | # | Frequency | %TOT |
|---|---|---|---|---|---|---|---|---|---|
| Ala | GCG | 29 | 0.15 | 0.73 | Pro | CCG | 16 | 0.12 | 0.40 |
| | **GCA** | 44 | 0.23 | 1.11 | | **CCA** | 36 | 0.27 | 0.91 |
| | GCT | 85 | **0.45** | 2.14 | | CCT | 58 | **0.43** | 1.46 |
| | GCC | 30 | 0.16 | 0.76 | | CCC | 24 | 0.18 | 0.61 |
| Cys | TGT | 94 | **0.76** | 2.37 | Gln | CAG | 25 | 0.44 | 0.63 |
| | **TGC** | 30 | 0.24 | 0.76 | | **CAA** | 32 | **0.56** | 0.81 |
| Asp | GAT | 54 | **0.66** | 1.36 | Arg | CGG | 23 | 0.31 | 0.58 |
| | **GAC** | 28 | 0.34 | 0.71 | | **CGA** | 21 | 0.28 | 0.53 |
| Glu | GAG | 87 | **0.6** | 2.19 | | CGT | 25 | **0.33** | 0.63 |
| | **GAA** | 58 | 0.4 | 1.46 | | CGC | 6 | 0.08 | 0.15 |
| Phe | TTT | 269 | **0.78** | 6.78 | Ser | AGG | 69 | 0.19 | 1.74 |
| | **TTC** | 78 | 0.22 | 1.97 | | **AGA** | 69 | 0.19 | 1.74 |
| Gly | GGG | 131 | **0.4** | 3.30 | | AGT | 55 | 0.15 | 1.39 |
| | **GGA** | 61 | 0.19 | 1.54 | | AGC | 23 | 0.06 | 0.58 |
| | GGT | 98 | 0.3 | 2.47 | | TCG | 18 | 0.05 | 0.45 |
| | GGC | 36 | 0.11 | 0.91 | | **TCA** | 33 | 0.09 | 0.83 |
| His | CAT | 37 | **0.62** | 0.93 | | TCT | 76 | **0.21** | 1.92 |
| | **CAC** | 23 | 0.38 | 0.58 | | TCC | 22 | 0.06 | 0.55 |
| Ile | ATT | 165 | **0.8** | 4.16 | Thr | ACG | 21 | 0.17 | 0.53 |
| | **ATC** | 40 | 0.2 | 1.01 | | **ACA** | 30 | 0.24 | 0.76 |
| Lys | AAG | 61 | 0.41 | 1.54 | | ACT | 57 | **0.46** | 1.44 |
| | **AAA** | 87 | **0.59** | 2.19 | | ACC | 16 | 0.13 | 0.40 |
| Leu | TTG | 122 | 0.23 | 3.08 | Val | GTG | 113 | 0.3 | 2.85 |
| | **TTA** | 210 | **0.39** | 5.29 | | **GTA** | 121 | **0.32** | 3.05 |
| | CTG | 43 | 0.08 | 1.08 | | GTT | 119 | 0.32 | 3.00 |
| | **CTA** | 70 | 0.13 | 1.76 | | GTC | 23 | 0.06 | 0.58 |
| | CTT | 75 | 0.14 | 1.89 | Trp | TGG | 58 | **0.54** | 1.46 |
| | CTC | 20 | 0.04 | 0.50 | | **TGA** | 49 | 0.46 | 1.24 |
| Met | **ATG** | 86 | 0.36 | 2.17 | Tyr | TAT | 103 | **0.69** | 2.60 |
| | ATA | 155 | **0.64** | 3.91 | | **TAC** | 47 | 0.31 | 1.18 |
| Asn | AAT | 76 | **0.66** | 1.92 | STOP | TAG | 34 | 0.58 | 0.86 |
| | **AAC** | 39 | 0.34 | 0.98 | | TAA | 25 | 0.42 | 0.63 |

**Note:**
The codons corresponding to a tRNA present in the mitochondrial genome are underlined and in bold. The highest frequency among synonymous codons is also underlined and in bold. #, number of codons; Frequency, frequency of each codon among synonymous codons; %TOT, frequency of each codon among all the codons.

pipeline; the complete input and output of this analysis is available on figshare (https://ndownloader.figshare.com/files/8360792). Figure 5 shows the variation in gene order between *Ruditapes decussatus* and *P. euglypta* (Fig. 5A), *M. lamarckii* F-type (Fig. 5B), *Ruditapes philippinarum* F-type (Fig. 5C), and among all the four species (Fig. 5D).

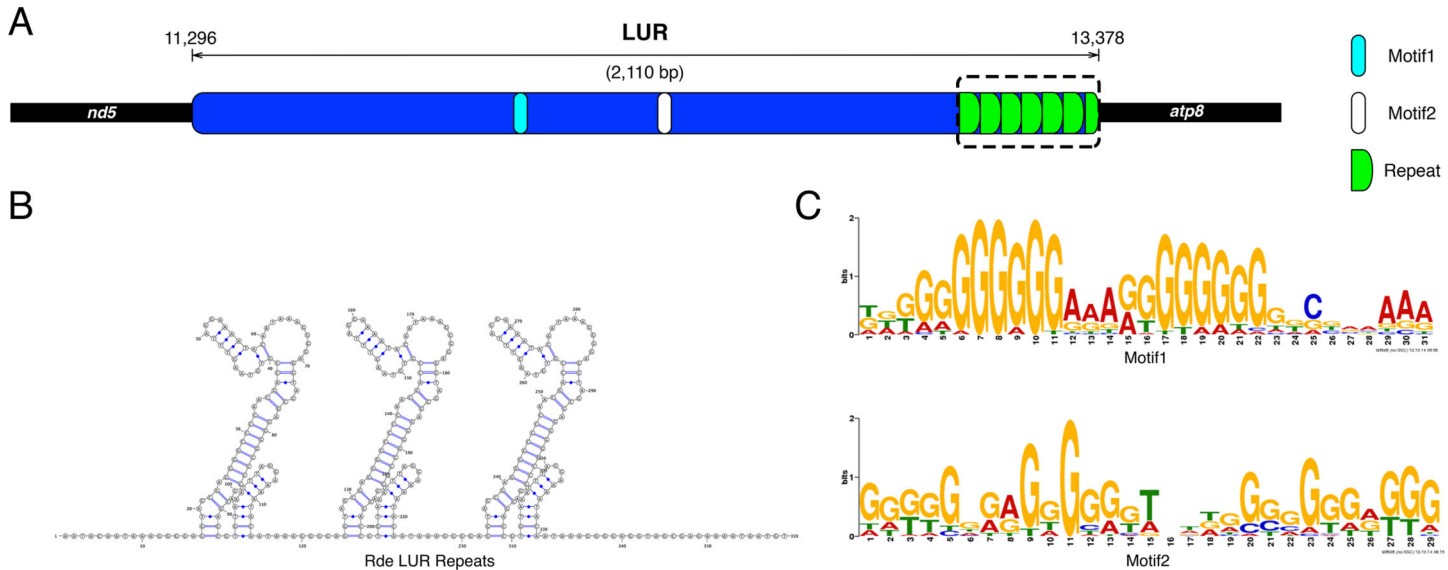

**Figure 2  Principal features of the Largest Unassigned Region (LUR).** (A): map of the lUR; (B): DNA secondary structure predicted in the repeat region (boxed in A); (C): Logos of the two DNA motifs found in the LUR.

Table 6  LUR length and number of repeats in the 13 female samples analyzed.

| Specimen | Length (bp) | Number of repeats | GenBank Acc. No. |
|---|---|---|---|
| F3 | 2,100–3,500 | 6.5–25 | MF055702 |
| F5 | 5,000 | 45 | MF055703 |
| F7 | 3,500 | 25 | MF055704 |
| F9 | 3,500 | 25 | MF055705 |
| F10 | 3,000 | 20 | MF055706 |
| F11 | 3,000 | 20 | MF055707 |
| F13 | 3,500 | 25 | MF055708 |
| F15 | 3,000 | 20 | MF055709 |
| F16 | 3,500 | 25 | MF055710 |
| F17 | 2,500–3,500 | 8–25 | MF055711 |
| F19 | 3,500 | 25 | MF055712 |
| F20 | 2,500 | 8 | MF055713 |
| F21 | 2,100 | 6.5 | MF055714 |

**Note:**
F3 and F17 are heteroplasmic with LURs of different length.

## DISCUSSION

### RNA-Seq-guided sequencing of mtDNA

The *de novo* assembly of the mtDNA from RNA-Seq data turned out to be informative, simplifying the primer walking procedure used for Sanger sequencing. Only one conting (Contig 1) resulted to be a chimeric sequence obtained by the misassembly of two smaller contigs. Most of the contigs (eight out of nine) contained more than one gene, and most of the tRNA genes were included in the *de novo* assembly. Except for *tRNA-Pro,*

**Table 7  Sequence Polymorphism (SP): SPs and small indels called by GATK.**

| Feature | Value | Min | Median | Mean | Max |
|---|---|---|---|---|---|
| Depth (all SPs) | – | 6 | 1,357 | 1,521 | 3,880 |
| Phred score (all SPs) | – | 3.30E+01 | 5.76E+03 | 4.18E+07 | 2.15E+09 |
| Depth (SPs in CDS) | – | 222 | 2,038 | 2,150 | 3,880 |
| Phred score (SPs in CDS) | – | 1.18E+02 | 1.01E+04 | 4.45E+07 | 2.15E+09 |
| Total number of SPs | 257 | – | – | – | – |
| Number of mRDI01 private SPs | 151 (58.7% of the total) | – | – | – | – |
| Number of SPs in CDS | 145 (56.4% of the total) | – | – | – | – |
| Number of mRDI01 private SPs in CDS | 103 (71% of the SP in CDS) | – | – | – | – |
| Number of SPs in CDS (excluding mRDI01) | 42 | – | – | – | – |
| Frequency of SPs in CDS | 0.0097 (~1 every 103 bp) | – | – | – | – |
| Frequency of SPs in CDS (excluding mRDI01) | 0.0028 (~1 every 355 bp) | – | – | – | – |
| Total number of indels | 18 | – | – | – | – |
| Number of indels in CDS | 6 | – | – | – | – |
| Number of indels causing frameshift | 4 | – | – | – | – |
| **# Of SPs** | **Whole mtDNA** | **CDS** | | | |
| Males | 234 (160) | 136 (107) | | | |
| Males (no mRDI01) | 84 (15) | 32 (6) | | | |
| Females | 97 (23) | 38 (9) | | | |

**Note:**
CDS, coding sequences; Whole mtDNA, polymorphism in the whole mitochondrial genome; the number in brackets the bottom of the table represent private SPs (e.g., there are 23 female specific SPs in the whole mtDNA and 9 female specific SPs in CDS); p-value, significance of the Fisher's exact test on number of SPs between sexes (i.e., all males vs females, males except mRDI01 vs females).

**Table 8  Indels located in coding sequences.**

| Position | Depth | Qual | Gene | SP | Frameshift | Sample | Allele frequency | Notes |
|---|---|---|---|---|---|---|---|---|
| 1,698 | 3,732 | 1.38E+04 | *cox1* | C/CAAA | No | mRDI02, mRDI03 | 0.089, 0.85 | Insertion of 1 Lysine |
| 6,364 | 1,929 | 2.15E+09 | *cytb* | CT/C | Yes | fRDI04, mRDI05 | 0.80, 0.81 | Yields a shorter *Cytb*. Possible sequencing error due to the homopolymer CTTTTTTT |
| 10,449 | 1,780 | 2.15E+09 | *nd5* | C/CT | Yes | fRDI01, fRDI04, fRDI05 | 0.11, 0.10, 0.11 | Yields a *nd5* gene divided in 2 ORFs. Possible sequencing error due to the homopolymer CTTTTTT |
| 17,619 | 2,272 | 5.98E+03 | *cox3* | AGCG/A | No | mRDI01 | 0.97 | Deletion of one Alanine |
| 17,621 | 2,188 | 9.99E+04 | *cox3* | CG/C | Yes | mRDI01 | 0.99 | Always combined with SP_17624. Together change the last 35 amino acids |
| 17,624 | 2,287 | 5.98E+03 | *cox3* | C/CAT | Yes | mRDI01 | 0.99 | Always combined with SP_17621. Together change the last 35 amino acids |

**Note:**
Depth, sequencing depth; Qual, quality of the called SP expressed in Phred score; Allele frequency, frequency of the alternative allele in each sample indicated in the "Sample" column.

*tRNA-Ile,* and *tRNA-Leu1*, all the other tRNA genes are organized in two big clusters: a 13-gene cluster positioned between *cox3* and *nd6*, and a 6-gene cluster between *nd6* and *nd4*. The assembly retrieved 6 out of 13 tRNAs from the first cluster (missing *tRNA-Gly, tRNA-*

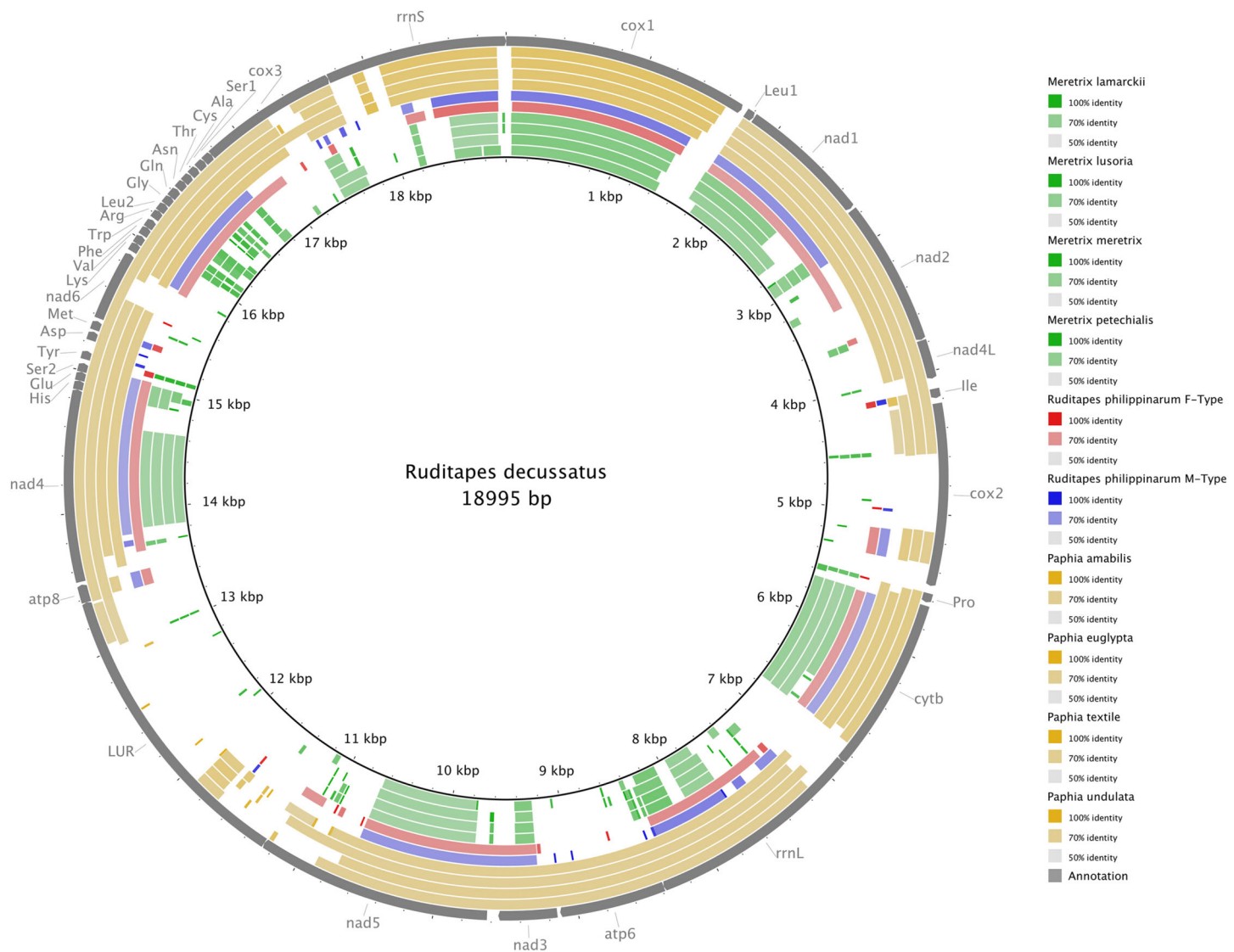

**Figure 3 BLASTN comparison of *Ruditapes decussatus* and other Veneridae.** *Ruditapes decussatus* mtDNA map (external gray circle), and BLASTN identity (colored inner circles) with complete mtDNAs of other 10 venerid species (see list in Table S3).

*Glu, tRNA-Asn, tRNA-Thr, tRNA-Cys, tRNA-Ala,* and *tRNA-Ser1*), and 4 out of 6 tRNAs from the second cluster (missing *tRNA-Met* and *tRNA-Asp*). All the tRNA genes not located in these two clusters (*tRNA-Pro, tRNA-Ile,* and *tRNA-Leu1*) were included in the contigs. The presence of a clear polyadenylation signal in four of the assembled contigs (see Table 1) seems to indicate the existence of multiple polycistronic transcripts. It is also noteworthy that poly-A sequences seem to be absent in contigs having tRNA of rRNA genes at one end (Contigs 2, 5, 8 and 9). This could be either an evidence supporting the "tRNA punctation model" of RNA processing proposed by *Ojala, Montoya & Attardi (1981)* for human mitochondria, or a result of difficulties in sequencing/assembly of such regions. More analyses are required to address this point.

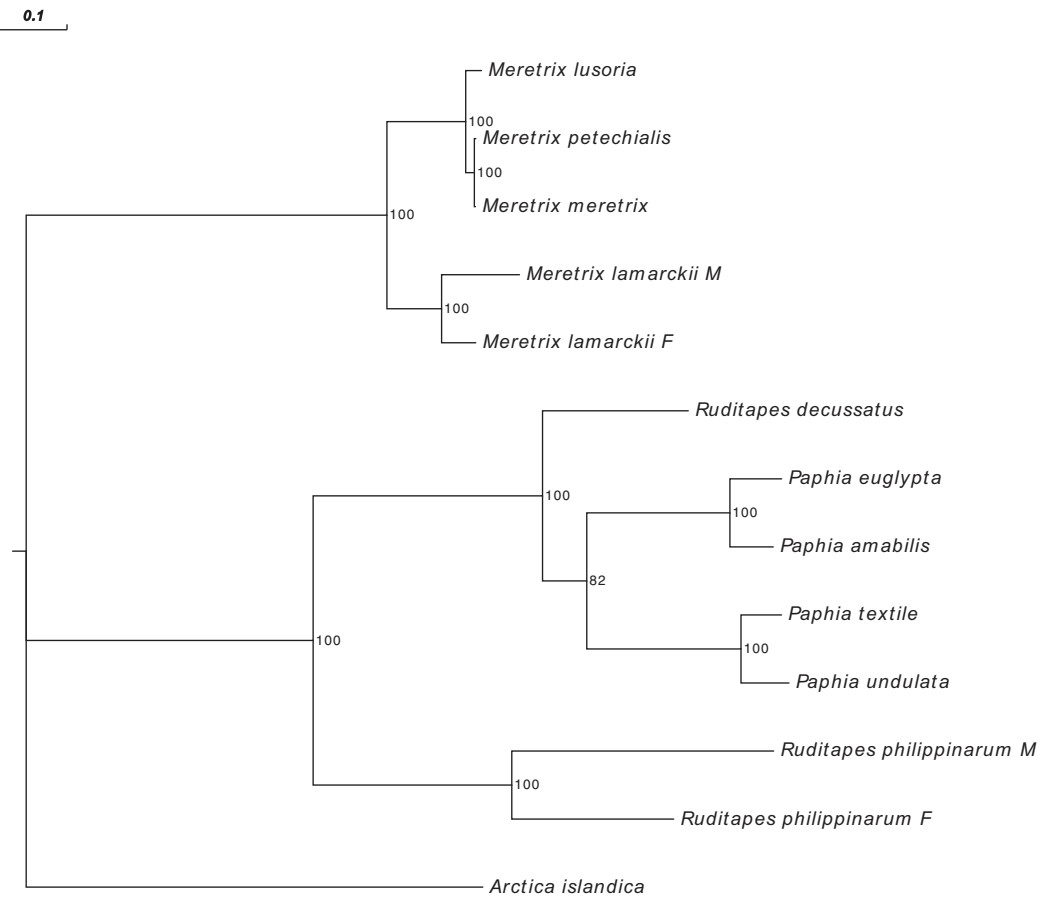

**Figure 4 Maximum Likelihood (ML) tree of Veneridae obtained with all mitochondrial coding genes.** ML tree obtained with the MitoPhast pipeline; the complete input and output of this analysis is available on figshare (https://doi.org/10.6084/m9.figshare.4970762.v1).

## General features

The size of the fully Sanger-sequenced mitochondrial genome of *Ruditapes decussatus* (reference female F4) is of 18,995 bp, and it includes 13 protein-coding genes, 22 tRNAs and 2 rRNAs. Our data support the presence of the *atp8* gene in the mtDNA of *Ruditapes decussatus; atp8* has been reported as missing in several bivalve species, however, more accurate searches often led to the identification of the gene, so, in most cases, the alleged lack of *atp8* is likely ascribable to annotation inaccuracies due to the extreme variability and the small size of the gene (*Breton, Stewart & Hoeh, 2010*; *Breton et al., 2014*; *Plazzi, Puccio & Passamonti, 2016*).

The mitochondrial genome of *Ruditapes decussatus* shows a high content of A-T (63%), a common feature in bivalve mtDNAs; moreover, T is the most common nucleotide at the third codon base (64.6%). The most common codon is UUU (Phe), which is also the most commonly used in bivalves, as well as in other invertebrates (*Passamonti et al., 2011*).

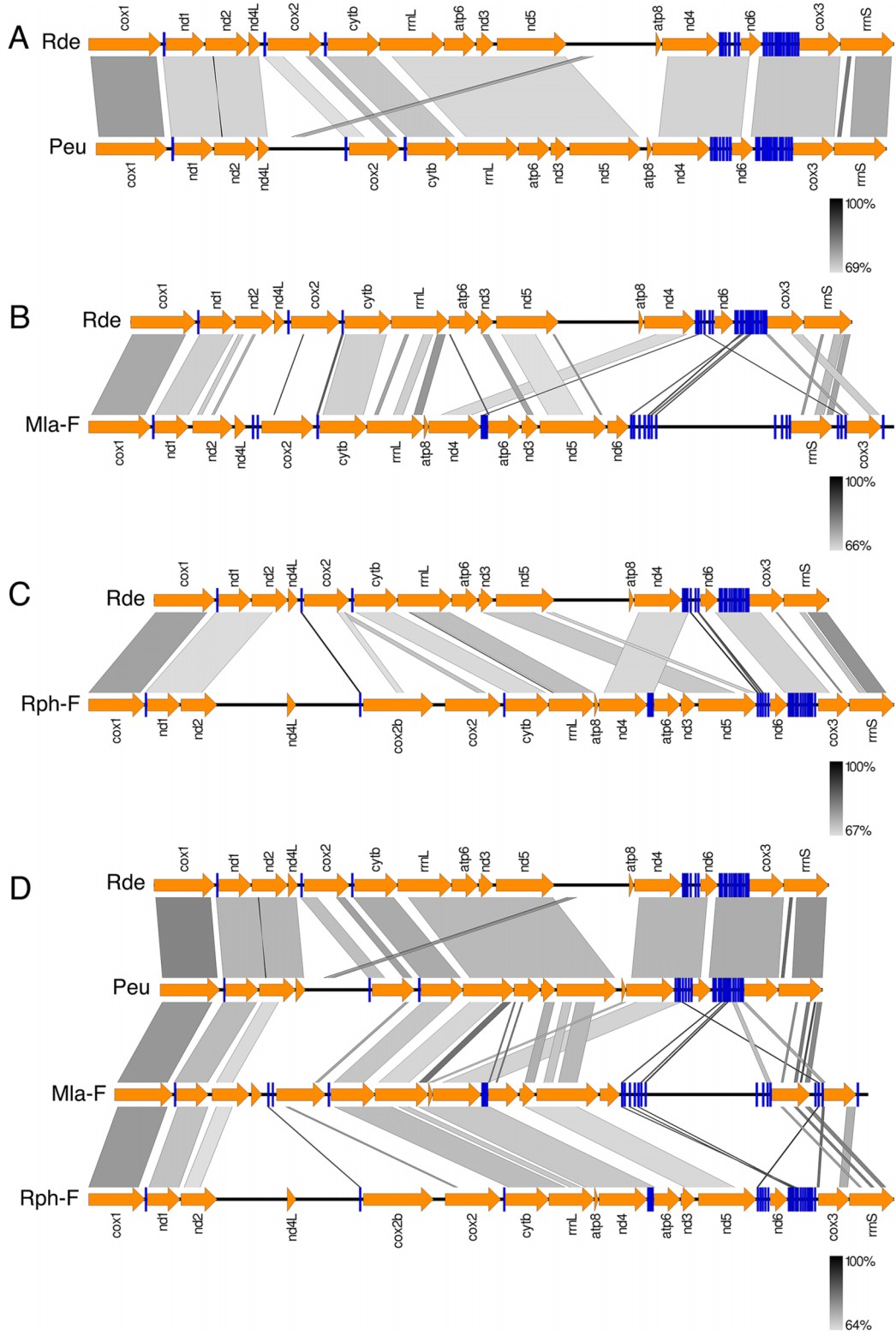

**Figure 5 Comparison of gene order in venerid mtDNAs.** Variation in gene order between *Ruditapes decussatus* and *P. euglypta* (A) *M. lamarckii* F-type (B) *Ruditapes philippinarum* F-type (C) and among all the four species (D).

## Codon usage

As shown in Table 5, in 16 cases out of 20, the most frequently used codon does not correspond to the anticodon of the inferred tRNA. In other words, there is not a correspondence between the most abundant codons and the anticodons of the 22 mitochondrial tRNAs. According to the "wobble hypothesis"—first proposed by *Crick (1966)*—the conformation of the tRNA anticodon loop enables some flexibility at the first base of the anticodon, so a Watson–Crick type of base pairing in the third position of the codon is not strictly necessary. This allows an amino acid to be correctly incorporated by ribosomes even if the tRNA is not fully complementary to the codon; according to Crick, this explains the degeneracy of the genetic code. This feature is particularly interesting in the light of the debate about natural selection acting at synonymous sites: since the early 1980s, evidence of a correlation between synonymous codon usage and tRNA abundances started accumulating. According to these authors, synonymous codon usage is biased to match skews in tRNA abundance, as a result of selective pressure maximizing protein synthesis rates (reviewed in *Chamary, Parmley & Hurst, 2006*). Following this rationale, the results here reported and data from other marine bivalves and metazoans (*Yu & Li, 2011*; *Passamonti et al., 2011*) would suggest that in some mitochondrial genomes translation efficiency is not maximized, and this observation deserves further investigation.

## Length and sequence polymorphism

The mtDNA of *Ruditapes decussatus* has a high proportion of URs mostly depending on the length of the LUR (Table 6); on average, bivalve mtDNAs have 1.7× the amount of URs in respect to other analyzed Metazoa (*Ghiselli et al., 2013*), and it is still unclear whether there is an accumulation of non-functional sequences in bivalve mtDNA due to genetic drift, or if such URs are maintained by natural selection because they contain—so far unknown—functional elements (see *Milani et al., 2013*, *2014b*; *Breton et al., 2014*; *Pozzi et al., 2017*). The LUR of *Ruditapes decussatus* most likely includes the mitochondrial CR, as indicated by the presence of two motifs (Fig. 2C; Figs S2 and S3) similar to two regulatory elements identified in the sea urchin CR. These two motifs are the same identified in previous analyses on the clam *Ruditapes philippinarum* and the mussel *Musculista senhousia* (*Ghiselli et al., 2013*; *Guerra, Ghiselli & Passamonti, 2014*) so they are conserved across distant bivalve taxa, and the GO terms associated with such motifs are related to transcriptional control (Table S4). An interesting feature of *Ruditapes decussatus* LUR is its variable length (Table 6), most likely due to different repeat content. As a matter of fact, the very same repeat sequence was present in every LUR, and our data strongly suggest that LUR length variation is actually due to repeat CNV (see Supplemental Information files on figshare: https://ndownloader.figshare.com/files/8387666 and https://ndownloader.figshare.com/files/8360789), as observed in other bivalve species (see *Ghiselli et al., 2013*; *Guerra, Ghiselli & Passamonti, 2014*). Tandem repeats have been also reported in the mitochondrial genomes of the bivalves *Acanthocardia tuberculata* (*Dreyer & Steiner, 2006*), *Placopecten magellanicus* (*La Roche et al., 1990*), *Moerella iridescens*, *Sanguinolaria olivacea*, *Semele scaba*, *Sinonovacula constricta*, *Solecurtus divaricatus*

(*Yuan et al., 2012*), *Ruditapes philippinarum* (*Ghiselli et al., 2013*), and *Musculista senhousia* (*Guerra, Ghiselli & Passamonti, 2014*). These repeats are believed to arise from duplications caused by replication slippage (*Buroker et al., 1990*; *Hayasaka, Ishida & Horai, 1991*; *Broughton & Dowling, 1994*). The tandem repeats found at the 3′ end of *Ruditapes decussatus* LUR are predicted to form a secondary structure (see Fig. 2B and Supplemental Information files on figshare) composed by multiple stem-loops, which obviously increase in number with the increment of the number of tandem repeats. The effect, if any, of tandem repeats in mtDNA is unknown: since the repeats are almost always localized in proximity of the CR, they might interact with regulatory elements—or even contain some—influencing replication and/or transcription initiation, and such interactions might also be altered by the formation of secondary structures (*Passamonti et al., 2011*; *Ghiselli et al., 2013*; *Guerra, Ghiselli & Passamonti, 2014*).

We assessed the genetic variability of *Ruditapes decussatus* mtDNA using two different approaches: by SP calling in CDS (RNA-Seq data on 12 individuals), and by analysis of the LUR (Sanger sequencing of 14 individuals). The CR and its flanking regions are known to be hypervariable, so they are commonly used to assess polymorphism at low taxonomic levels. Our data strongly support a very low genetic variability: the number of SPs in CDS is 145, of which 103 are private of a single individual (mRDI01)—thus reducing the number to 42—while the number of variable sites in the analyzed LURs is 98 over 3,095 aligned positions. Considering the known variability of mtDNA in bivalves (*Gissi, Iannelli & Pesole, 2008*; *Ghiselli et al., 2013*; *Breton et al., 2014*; *Plazzi, Puccio & Passamonti, 2016*), this is a surprising result. Even more if we compare the results of the present work to a methodologically identical analysis performed on 12 *Ruditapes philippinarum* samples from the Pacific coast of USA, performed by *Ghiselli et al. (2013)*: in that work, GATK yielded 194 SPs in the M-type mtDNA and 293 in the F-type. Strikingly, the 12 *Ruditapes philippinarum* samples analyzed were actually two families (6 siblings + 6 siblings). This means that randomly sampled individuals of *Ruditapes decussatus* used in this work showed a much lower mtDNA variability than *Ruditapes philippinarum* siblings. A previous analysis on the *cox1* gene of *Ruditapes decussatus* reported a nucleotide diversity (π) of 0.15 for a population from the Northern Adriatic Sea (*Cordero, Peña & Saavedra, 2014*). Another analysis on the same gene of *Ruditapes philippinarum* from the same range resulted in a π = 0.25 (*Cordero et al., 2017*), so *Ruditapes decussatus* has a lower nucleotide diversity at the *cox1* locus. The difference between the variability in mtDNA of *Ruditapes decussatus* that we are reporting here and that of *Ruditapes philippinarum* reported in *Ghiselli et al. (2013)* appears to be more marked. It is known that the genetic variability of *Ruditapes philippinarum* in the Adriatic Sea is lower than in populations from its native range in Asia (*Cordero et al., 2017*), probably because of the bottlenecks that this species had to go through during the multiple colonization events. The introduction in North America from Asia happened first (in the 1930s), and from there the Manila clam was introduced in Atlantic Europe (in the 1970s and 1980s), and lastly into the Adriatic Sea (1983 and 1984), and it is plausible that the genetic diversity decreased at each introduction event. Accordingly, *Cordero et al. (2017)* observed that *Ruditapes philippinarum* genetic variability in Europe is lower compared to that of the Pacific coast

of the USA, so the samples analyzed in *Ghiselli et al. (2013)* could have been more polymorphic than those analyzed in *Cordero, Peña & Saavedra (2014)*, thus explaining the more pronounced differences in genetic variability between the Manila clam and the European clam discussed above. In any case, all the available data point to a lower genetic diversity of *Ruditapes decussatus* mtDNA, and it would be interesting to know whether it is a cause or an effect of the ongoing replacement of *Ruditapes decussatus* with the invasive *Ruditapes philippinarum*. It will also be important to investigate genetic variability of the nuclear genes, especially after *Cordero, Peña & Saavedra (2014)* reported contrasting levels of differentiation between mitochondrial and nuclear markers.

With respect to SP effects, we found six indels in CDS, 2 of which do not cause frameshift, but a simple insertion/deletion of one amino acid (SP_1698, and SP_17619, see Table 8). Of the remaining four, SP_6364 and SP_10449 consist of a deletion and an insertion of a single T in two homopolymeric sequences (CTTTTTTT and CTTTTTT, respectively), raising the possibility of a sequencing error. In any case, the two SPs yield a shorter CDS (*cytb* and *nd5*, respectively), and are present at relatively low frequencies in the specimens carrying them, except for SP_6364 which has a frequency of 80% in fRDI04. The *cox3* gene shows three SPs: the first one, SP_17619, does not cause a frameshift, and results in the deletion of one alanine residue, and its frequency in mRDI01 is 97%. The second one, SP_17621, consists of a deletion of a G with respect to the reference sequence, which is the Sanger-sequenced mtDNA of sample F4; all the individuals analyzed with RNA-Seq carry this deletion except for mRDI01 which, at that position, has the same sequence of the reference mtDNA (reference-like allele frequency in mRDI01 = 99%). The third indel, SP_17624, consists of an insertion of two nucleotides, and its frequency in mRDI01 is 99%. So, basically, for *cox3* we have three types of sequences: (i) the Sanger-sequenced reference, which yields a 966 bp (321 aa) ORF; (ii) a sequence found in 11/12 of samples analyzed with RNA-Seq (except mRDI01) that carries a single-nucleotide deletion (SP_17621), and yields a 963 bp (320 aa) ORF; (iii) a sequence, private of mRDI01, which is obtained by combining SP_17624 and SP_17621 (both 99% of frequency, so most likely co-occurring), which produces a 963 bp (320 aa) ORF. Interestingly, the ORFs obtained from the sequences described in (ii) and (iii), are almost identical, namely the sequence obtained by RNA-seq in 11/12 samples and the sequence obtained by RNA-Seq in mRDI01 are basically the same, and differ from the Sanger-sequenced reference, yielding an amino acid sequence that differs in the last 35 residues (all data available in Supplemental Information files on figshare: https://doi.org/10.6084/m9.figshare.4970762.v3). Given this consistent difference between the sequence obtained by Sanger-sequencing of DNA, and those obtained by RNA-Seq, it is tempting to speculate that this difference might be caused by RNA editing, a mechanism observed in mtDNA of some animals (*Lavrov & Pett, 2016*), and recently reported to be common in cephalopods (*Liscovitch-Brauer et al., 2017*). Actually, *Liscovitch-Brauer et al. (2017)* reported only A-to-I editing, which is not the kind of change we are observing here, but other types of editing are known across eukaryotes (see *Gott & Emeson, 2000* for a review), and some others, still unknown, might exist as well. Post-transcriptional modifications (thus including RNA-editing) are still poorly understood mechanisms, but they appear to be responsible for

most of the mitochondrial gene expression regulation (*Scheibye-Alsing et al., 2007*; *Scheffler, 2008*; *Milani et al., 2014a*). What we propose here is a pure conjecture, but we think in the future it might be worthy to investigate mitochondrial transcriptomes looking for such kind of "unexpected" biological features.

Interestingly, in contrast with a low nucleotide variability along the entire mitochondrial genome, we observed a pretty high polymorphism in LUR length due to CNV of tandem repeats, and even a LUR length heteroplasmy: two females yielded two electrophoretic bands each (~2,100 and ~3,500 bp in F3; ~2,500 and ~3,500 bp in F17; see Table 6). A possible explanation is that the diversity (CNV) detected in the LURs could be recent: the accumulation of nucleotide variation at different sites along the mitochondrial genome needs time, while the kind structural variability we observed can be achieved in few generations (or even one) considering that replication slippage is common in repeat-rich regions.

## Phylogenetic relationship with *Ruditapes philippinarum*

Despite *Ruditapes decussatus* and *Ruditapes philippinarum* being morphologically similar and being ascribed to the same genus, the results here reported clearly show that they are quite different both for mtDNA sequence (Figs. 3 and 4) and mtDNA gene arrangement (Fig. 5). This is an unusual finding, even among bivalves, which are known to be fast-evolving for these characters. This may point to the fact that these two species are less related than previously thought. Actually, this is not the first clue that *Ruditapes decussatus* and *Ruditapes philippinarum* are quite different genetically, as allozyme electrophoresis (*Passamonti, Mantovani & Scali, 1997*, *1999*) and satellite DNA content (*Passamonti, Mantovani & Scali, 1998*) pointed out. More in-depth analyses are therefore needed to correctly trace the phylogenetic relationships of these two Ruditapes species, which may eventually end up in two different Genera. As shown in Figs. 3–5, the Genus Paphia is the most similar to *Ruditapes decussatus*.

## Presence/absence of DUI

We could not find evidence for sex-specific mtDNAs, typical of DUI. As stated in the Introduction, the search for DUI is not a straightforward process. HTS can help thanks to a much deeper sequencing coverage (in respect to the cloning-and-Sanger-sequencing approach), and because it overcomes the problem of primer specificity, a limitation of the classical approach. One possible concern about using HTS approaches based on short reads in presence of DUI is about the ability of softwares to detect divergent reads and assembly them correctly. More specifically, one could ask what is the divergence threshold under which the assemblers are not able to partition the contigs into two sex-linked groups. We do not know such a threshold, but we used different assembly strategies trying to retrieve sex-specific mtDNA sequences from our data. Other than the approach reported in Materials and Methods (which is the one that produced the data reported here), we tried other techniques. After identifying reads that blasted to bivalve mitochondrial sequences present in GenBank and discarding all the other reads, we generated A5+CAP3 assemblies: (i) for each of the samples (obtaining 12 separate assemblies), and (ii) pooling the six males together and the six females together, and

assembling the two sex-specific pools. Both these approaches did not show evidence of sex-specific mtDNAs. Then we took the assembly obtained from the females and removed the reads from each of the samples that mapped (<8 mismatches) to these sequences. We then used the remaining reads as A5 input. The program could not assemble anything. Lastly, we tried the software MetaVelvet (*Namiki et al., 2012*)—that assembles metagenomes—on all the reads matching bivalve mtDNAs, and only one genome was produced. After all these alternative approaches failed to find two sex-linked mtDNAs, we decided to proceed with the assembly as indicated in Materials and Methods, because it was the technique that yielded the best quality contigs, most likely because using the reads from all 12 the individuals granted a higher coverage of the mtDNA. Given these results, we can propose three different explanations.

1. *Ruditapes decussatus* is characterized by SMI of mitochondria, so a male-transmitted mtDNA is not present in this species.

2. The divergence between the two sex-specific mtDNAs is too low to be detected. This could be the outcome of two different situations.

    a) DUI is very young in this species, so the two sex-linked mtDNAs did not have the time to diverge.

    b) A role-reversal event occurred recently. Role reversal (a.k.a. "route reversal" or "masculinization") is a process—observed so far only in species of the Mytilus complex—by which F-type genomes invades the male germ line becoming sperm-transmitted, thus turning into M-type mtDNAs (*Hoeh et al., 1997*). This event actually resets to zero the divergence between F- and M-type, although substantial differences in the control regions were reported between the original F-type and the "masculinized" one (see *Zouros, 2013* for a thorough review). The hypothesis that role reversal could have occurred multiple times in the evolutionary history of bivalves and could have led to the complete replacement of M or F mtDNAs in several species was proposed by *Hoeh et al. (1997)* to explain the scattered phylogenetic distribution of DUI across Bivalvia. Indeed, according to the hypothesis of a single origin, DUI arose >400 Mya, approximately at the origin of Autolamellibranchia, but, as said, such hypothesis requires the assumption of multiple role-reversal and/or DUI loss events in several branches of the bivalve tree (see *Zouros, 2013* for a detailed discussion). Recently, a multiple origin of DUI was proposed (*Milani et al., 2013*, *2014b*; *Milani, Ghiselli & Passamonti, 2016*; *Mitchell et al., 2016*), and in such case there would be no need of multiple role-reversal events to explain its phylogeny. In our opinion, until further evidence will be provided, role-reversal should not be considered a rule, but rather an exception. Of course, we cannot rule out that a masculinization event might have occurred in *Ruditapes decussatus*, so this hypothesis must be taken into consideration.

3. In our data, even if there is no clear evidence of a male-specific mtDNA, a male sample (mRDI01) clearly stood out from the others, both males and females (see Table 7). Overall, the divergence between mRDI01 and the other 11 samples calculated

considering its private SPs is of 151 sites over 18,995 bp (considering the whole mtDNA), and of 103 sites over 14,920 bp (considering only CDS). In both cases the divergence is very low (0.8% and 0.7%, respectively), which explains why the mtDNA of mRDI01, although different, was not assembled as a separate genome. We have no sufficient data to evaluate if such divergence is normal within *Ruditapes decussatus* populations, but considered the variability usually observed in bivalves, we find the difference unsurprising. On the contrary, the lack of variability among the other 11 samples is remarkable. For these reasons, we are inclined to believe that mRDI01 divergence is compatible with hypotheses (1) and (2). That said, there still could be a third, quite conjectural, hypothesis by which these data might indicate an incipient DUI, not yet fixed in the population.

All in all, we have a preference for the first explanation, but the present data are not sufficient to exclude the others, and a more thorough investigation is necessary to assess this point.

Up to now DUI was identified in only three Veneridae species: *Cyclina sinensis*, *Ruditapes philippinarum*, and *Meretrix lamarckii* (*Gusman et al., 2016*). The status of Paphia is still unknown, and in future works it would be interesting to investigate more Heterodonta species to understand better the distribution of DUI in this derived group of bivalves.

## ACKNOWLEDGEMENTS

We would like to thank Edoardo Turolla (Istituto Delta Ecologia Applicata, Ferrara, Italy) for providing the specimens, and Massimo Milan for bibliographic suggestions. We also gratefully thank the Editor Tim Collins, and the reviewers Carlos Saavedra, Shallee Page, and one anonymous colleague for their comments and suggestions.

### Funding

This study was supported by the Italian Ministry of Education, University and Research (MIUR) FIR Programme no. RBFR13T97A funded to FG, MIUR SIR Programme no. RBSI14G0P5 funded to LM, Zumberge Foundation to SVN, and by the Canziani bequest funded to MP. The funders had no role in study design, data collection and analysis, decision to publish, or preparation of the manuscript.

### Grant Disclosures

The following grant information was disclosed by the authors:
Italian Ministry of Education, University and Research (MIUR) FIR: RBFR13T97A.
MIUR SIR: RBSI14G0P5.
Zumberge Foundation.
Canziani bequest.

## Competing Interests

The authors declare that they have no competing interests.

## Author Contributions

- Fabrizio Ghiselli conceived and designed the experiments, performed the experiments, analyzed the data, wrote the paper, prepared figures and/or tables, reviewed drafts of the paper.
- Liliana Milani conceived and designed the experiments, performed the experiments, analyzed the data, prepared figures and/or tables, reviewed drafts of the paper.
- Mariangela Iannello performed the experiments, analyzed the data, prepared figures and/or tables, reviewed drafts of the paper.
- Emanuele Procopio performed the experiments, analyzed the data.
- Peter L. Chang analyzed the data.
- Sergey V. Nuzhdin conceived and designed the experiments, contributed reagents/materials/analysis tools, reviewed drafts of the paper.
- Marco Passamonti conceived and designed the experiments, contributed reagents/materials/analysis tools, reviewed drafts of the paper.

## DNA Deposition

The following information was supplied regarding the deposition of DNA sequences:
   GenBank accession numbers MF055702 to MF055714, and KP089983.
   GenBank BioProject PRJNA170478.

## Data Availability

   Ghiselli, Fabrizio; Milani, Liliana; Iannello, Mariangela; Procopio, Emanuele; L. Chang, Peter; Nuzhdin, Sergey; Passamonti, Marco (2017): The Complete Mitochondrial Genome of the Grooved Carpet Shell, *Ruditapes decussatus* (Bivalvia, Veneridae). figshare.
   https://doi.org/10.6084/m9.figshare.4970762.v3.

## Supplemental Information

Supplemental information for this article can be found online at http://dx.doi.org/10.7717/peerj.3692#supplemental-information.

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
