# Peer review of "The complete mitochondrial genome of the grooved carpet shell, Ruditapes decussatus (Bivalvia, Veneridae)"

_PeerJ, doi:10.7717/peerj.3692_

## Round 0.1 · original submission · Major Revisions

In this manuscript, the authors describe the mitochondrial genome of the bivalve Ruditapes decussatus. The genome was sequenced in a two-step process with a first draft generated from RNA-seq data, and this draft genome subsequently verified by primer-walking Sanger sequencing. Significantly, they authors conclude that they find no evidence of Doubly Uniparental Inheritance (DUI), common in bivalves, with no confirmation of the male haplotype, typically widely divergent from the female haplotype.

Three external reviewers provided detailed and useful comments. They found the manuscript to be solid overall, but had questions that will require significant revision. One important point concerned that inference of no DUI, with reviewer 1 suggesting the possibility that the female haplotype may have relatively recently displaced the male haplotype, as has been found in some other bivalves. Similarly, reviewer 3 suggests that, depending on the state of the gonads, male haplotypes could be quite rare, and proposes that sampling of sperm may be the most definitive way to capture the male haplotype, if present. Along those lines, I wonder if the single male haplotype responsible for 71% of all of the SNPs found in the coding sequences might represent a somewhat diverged male haplotype resulting from a past "masculinization" or role reversal event. In your phylogeny, in both cases where you have both male and female haplotypes from the same species, the within-species male and female haplotypes are sister taxa. This is what you would expect if role reversals were happening with some frequency in the clade. If, in contrast, male and female lineages were maintaining their integrity over long time spans, you would expect parallel clades of all male or all female haplotypes. Finally, reviewer 2 felt that more detail needed to be provided regarding analyses. I agree, particularly about the RNA-seq analysis and primer-walking, since these are novel aspects of the approach outlined in the study. Was the RNA-seq assembly complete, or in several pieces? How many fragments were used for the primer walking verification?

I have provided additional comments and editorial suggestions. The Editorial Office will send you the Word version of the file and I append a PDF version of it here, for the record.

·

Basic reporting

1.1. - In lines 298-300 the following sentence can be read: " GATK output in VCF format and a detailed list of SNPs in tabular format is available on Figshare: https://doi.org/10.6084/m9.figshare.4970762.v1";. The link leads to the Figshare web page where four zip and one txt files are available. However, none of them corresponds to the mentioned GATK output.

1.2. Fig 1. This figure shows a map of the genes found in the sequenced mtDNA. Authors should consider whether the positions of the unassigned regions (at least the largest one, LUR) should be indicated.

1.3. - The database accessions of the sequences obtained in this work are missing, with the exception of the sequences of the LUR region. Authors should not forget to include them in the revised version.

Experimental design

No comment

Validity of the findings

3.1. - The authors have established that R. decussatus does not show sex-linked mitochondrial genomes. They interpret this result as an evidence for the absence of the mitochondrial DNA inheritance system called DUI that has been found in other clams of the same family and in other bivalves. The data they have gathered support this conclusion. However, there is the possibility that they are confronting a phenomenon that has been described in mussels of the genus Mytilus, which also show DUI. In this genus, some paternally-transmitted, M genomes have been found which show a sequence which is phylogenetically closer, sometimes almost identical, to the maternally transmitted, F genomes. The phenomenon is known as "masculinization" or "role reversal", and it has been explained as the result of one F genome being "accidentally" captured by the paternal transmition route, and spreading through the population (e.g.: Hoeh, W. R., D. T. Stewart, C. Saavedra, B. W. Sutherland and E. Zouros. 1997. Phylogenetic evidence for role-reversals of gender-associated mitochondrial DNA in Mytilus (Bivalvia: Mytilidae). Mol. Biol. Evol. 14(9): 959-67). Phylogenetic evidence has been obtained that suggest that role-reversals could have occurred several times in the evolutionary history of bivalves and could have led to the complete replacement of a species' M or F genome (e.g.: Hoeh, W. R., D. T. Stewart, B. W. Sutherland and E. Zouros. 1996. Multiple origins of gender-associated mitochondrial DNA lineages in bivalves (Mollusca: Bivalvia). Evolution. 50(6): 2276-2286). A very recent masculinization event in R. decussatus could limit the possibility to distinguish F and M genomes in this species. The authors should consider if this possibility is applicable to their study, and deal with it in the Discussion.

Additional comments

- Line 402- Consider substituting Northern by Atlantic

·

Basic reporting

Summary: This manuscript is a solid analysis of the mitochondrial sequence and structure of Ruditapes decussatus. It is an interesting and worthwhile contribution. Only minor revision is recommended.

1. Basic reporting
The writing is clear and the literature review is quite complete. I have made a few small suggestions listed below.

Abstract and Introduction - Repeated word ‘sketchy’. I don’t disagree with the characterization but it would be more descriptive to say that the diversity is not well sampled, and that even basic understanding, such as the biological basis for gene order and the evolutionary implications of mitochondrial inheritance are lacking.
“The unavoidable consequence was the replacement of R. decussatus with R. philippinarum”. Delete ‘unavoidable’. I’d argue that the replacement was avoidable. It was only the consequence of the choices made in aquaculture. I expect the authors mean that, following introduction of the Manila clam, the supplanting of R. decussatus was unavoidable. Or that upon introduction of the more robust species, replacement was inevitable.

Experimental design

2. Experimental design
The experimental design is appropriate and consistent with other publications on mollusk mitogenomes. Although this is a fairly standard mitogenome analysis, this is an important species to understand, and the authors provide useful analysis that adds to the body of literature. It seems to fit easily within the Aim and Scope of the journal.

The Introduction refers to the need for a “more widespread approach”, but their approach is far from high-throughput---though their sample size is better than most. Notably, the use of transcriptome as a starting point for genome analysis is less common and timely. Using RNAseq and then performing confirmation by Sanger sequencing is a welcome approach that complements most mitogenome papers in the literature. Likewise, the sexing of the individuals is appreciated, so that one can have data regarding the DUI and heteroplasmy. I would like more experimental detail, particularly about software used and parameter selection, as listed below.
I think that the RNAseq data and the gender differences could be and should be more fully explored, but it could be more appropriate to form the basis of a separate manuscript.
Previously unappreciated functionality such as use of sncRNAs are being rapidly explored. The putative RNA editing is fascinating, though highly speculative. A bit more analysis of the flanking sequences might shore up their arguments. Understandably, the authors might be more interested in staking a claim to the idea of mitochondrial RNA editing in the bivalves than making a case for it. The abstract notes that the RNA editing is putative, but I would like to see an acknowledgement in the Discussion that it is in fact quite speculative.

Validity of the findings

3. Validity of the findings
With the exception of the RNA editing idea, the findings are solid, the figures are well-done and the discussion appropriate.

Line-by-line annotations and suggestions:

Line 144 – were all individuals harvested from the same location at the same time? Are GPS coordinates included in the GenBank record? [I can’t pull up any of the data from GenBank listed in Supplemental Table 1, not even the Sanger sequencing(# KP089983). Has GenBank only given tentative assignment of accession numbers?]
Line 165 - awkwardly transferred from Tritt et al abstract.
Reword. e.g,. "an assembly pipeline with automated progression through the following stages.."
Line 166 – Reword: The de novo assembly pipeline was used to analyze our FASTQ Illumina data using default parameters.
Line 168 - Was this used from ngopt? Using the original version of the software as of 2012?
Line 169 – QC score cutoffs?
Line 169 - Under Methods, I'd strongly prefer a statement of what was done rather than what the programs does or what it can do.
Line 175 - Whole organism or gonadic tissue was frozen?
Line 181- F4 designation struck me at first as a generation nomenclature, rather than 'female individual'. The identification of the individual sequenced is already listed in Supp. Table 1, so I'd delete that parenthetical designator for clarity.
Line 210-218 - The LURs in the list of species are not terribly concordant. Were some the LUR identifications in some species privileged in the analysis? Or was the identification done by the motifs with the greatest amount of species contributing to that as a consensus?
Line 237 – What was the basis for choosing ML? Was Bayesian inference considered? (Not that Fig 4 is not completely plausible. The tree looks reasonable).
Line 248-9 – I’d like more explanation here or in Discussion. There is no reference to this search in the Methods that I can find. The phrasing “were not retrieved” implies that specific sequences were used to fish for the gender haplotypes, but I don’t see them referenced. Or was the search done by separating the sequences of the two genders in pools and searching for sequence variablility de novo? Either method is potentially acceptable but it needs to be specified and defended.
Some discussion comparing this result to the literature would be appropriate. It is not terribly surprising that sex-linked sequences were not obtained, but the result should be explained and put into context.
Were multiple MT genes examined? Was rrnL used? I’m not terribly surprised by the MSA result, given the high nucleotide divergence in the mollusks. But, a little more discussion would be welcome.
I also wonder if overall variability was measured between male and female. There is some indication that male-inherited DNA evolves more quickly.
The Clustal Files opened fine in Clustal Omega, but I had to search for them a bit in FigShare. Direct file links would have been more convenient than having to dig through the cryptic folders in FIgShare from a single URL from Supplemental Table 1.
Line 245- Was Fig 5 generated with Artemis Comparison Tool? Not listed in Methods.
Line 251- What mollusks have the closest gene order? The ones listed? Or were those chosen based on phylogeny?
Line 276- typo. Word missing?
The tripartite structure of the LUR was nicely captured.
Line 297 – One wonders if mRDI01 has cancer….
Line 308- Fig 5 – Is there any meaning to whether the lines are in front of or behind each other in the gene order diagram?
Line 322-325 – I am not sure that this statement about ATP8 is relevant since you only are analyzing one species. Furthermore, if anything, the ATP8 in your Figure 3 seems to support the opposite conclusion. Breton has made the same statement but no one seems interested in going back to those previously sequenced mitogenomes and demonstrating that it is correct.
Line 341-346 – The analysis here seems fine. Rather than only utilizing Metazoa, there are a couple of papers examining this question closely in mollusks. [E.g., Yu & Li, 2011. PLoS One]. At least one of these should be referenced.
354-355 – Kudos for pulling up a putative CR based on urchin!
Should Table 1 go to supplemental figures? Actually, I wonder if Editor would like to see several of the Tables go to Supplemental, especially the last one, the gene ontology table that essentially just says ‘transcriptional regulation’.
Supplemental table 3 is a very convenient list.
Table 7 – I was surprised at the lower Phred scores for SNPs. Were all of the SNPs scored or was there a lower sequencing QC cutoff? Methods don’t list the sequencing QC parameters.
Line 383 – Only 42 SNPs? As the authors note, such low nt divergence seems very surprising. It makes me particularly interested in more detail about the organism harvesting method and location, not found in Methods.
And, the clustal alignment in FIgShare is the MSA of that LUR, correct? That variable region MSA looks typical of the bivalves, but I wonder if the % identity in the LUR is also low.
Line 429 – I find the RNA editing supposition unconvincing. I have read of uracil deletion. Is there literature precedent for guanine deletion? Does the surrounding sequence resemble consensus for programmed frameshift (esp. poly-G containing)? Luscavitch reports RNA editing of an A-to-I, not a deletion.
I am surprised that this is the only use of the complete transcriptomes derived from the RNAseq. Save for this, the paper is a solid analysis that could have been done by mitogenome DNA sequencing.
Line 437 typo
Line 449 – This explanation is very plausible.
Line 461 – also, previous phylogenetic analyses (based on a few MT genes and a nuclear gene) bear out this muddle with Paphia: see Kappner, 2006. Molecular Phylogenetics and Evolution 40(2):317-31.
I might like to see a short paragraph at the end putting these findings into a broader context of bivalves.

Reviewer 3 ·

Basic reporting

No comment.

Experimental design

Did the authors try to spawn the mussels? DNA isolation from spawned sperm would be the most definitive way to confirm if DUI is present in this species or not. Even better, to put it through Percoll as per Venetis, Constantinos, et al. "No evidence for presence of maternal mitochondrial DNA in the sperm of Mytilus galloprovincialis males." Proceedings of the Royal Society of London B: Biological Sciences 273.1600 (2006): 2483-2489.
I think that a more thorough description of the reproductive tissue/structures of this mussel coupled with an assessment of the degree of sperm development at the time of dissection would help evaluate whether the approach used could have heavily biased the sample in favor of female transmitted mtDNA in the event that DUI is actually present. How “ripe” were the gonads? Was DNA from the entire animal analyzed (which would bias it in favor of F type in males with DUI) or were only gonad samples used? Even if gonad samples were used, depending n their state of maturation/development, they could be heavily dominated by F-type mtDNA.
Does the De novo assembly approach used produce only one dominant assembled genome even if two genomes are present? Have the authors also considered whether they might be looking at an extremely recently masculinized heteroplasmic male? How would the assembly process treat that case?

Validity of the findings

The validity of some aspects of the results and interpretation will rely on answers to the questions posed above in Section 2.

Additional comments

Review of the complete mitochondrial genome of the grooved carpet shell, Ruditapes decussatus (Bivalvia, Veneridae) (#18041) by Ghiselli et al.

This is a straightforward analysis of several aspects of the mitochondrial genome sequence of this bivalve mollusk. The authors use a combination of RNA-Seq data and direct Sanger sequencing to generate the sequence of this genome. The methods used are appropriate, and generally well-described and justified, although I do think that further discussion of potential biases in various aspects of the analysis may be relevant for detecting a potential male-transmitted genome in this organism. (I will discuss this further in a moment.)
The manuscript is well written and the figures are all legible and helpful.
I have the following editorial suggestions:
l. 26. Replace “sketchy” with “relatively limited”. As written, sketchy suggests that the previous analyses conducted are of dubious or questionable quality, which is not what I think you mean to say.
l. 37. Add that CNV means “copy-number variation”
l. 46. Should read. “Strikingly, in contrast with …”
l. 47. Should read “…observed a high degree of polymorphism…”
l. 67. Re4place “sketchy” as above.
l. 95. Review what you mean by the statement “currently the production of the the European clam is almost insignificant” in more detail. Is R. decussatus now endangered? Threatened? Is that what you are implying?
l. 100-101. Do you mean proportion or number of URs? Clarify.
l. 107. I suggest replacing with “…feature of bivalve mitochondrial DNA is the Doubly Uniparental Inheritance (DUI) system of transmission…”
l. 120. Add “…despite the fact that many…
l. 132. Add a reference for the statement that for HTS applications that low copy-variants can be easily unveiled. (Also, I would substitute some other word for “unveiled”.) This will be important for evaluation how likely it might be that you could miss obtaining sequence for, and hence evidence of, a male-transmitted mtDNA genome using this technique.
l. 143. You mentioned that samples were collected during the spawning season, but when (in terms of dates) was this actually done. These data are not included in the supplementary files as far as I could see.
l. 143. Did you try to spawn the mussels? DNA isolation from spawned sperm would be the most definitive way to confirm if DUI is present in this species or not. Even better, to put it through Percoll. Venetis, Constantinos, et al. "No evidence for presence of maternal mitochondrial DNA in the sperm of Mytilus galloprovincialis males." Proceedings of the Royal Society of London B: Biological Sciences 273.1600 (2006): 2483-2489.
I think that a more thorough description of the reproductive tissue/structures of this mussel coupled with an assessment of the degree of sperm development at the time of dissection would help evaluate whether the approach used could have heavily biased the sample in favor of female transmitted mtDNA in the event that DUI is actually present. How “ripe” were the gonads? Was DNA from the entire animal analyzed (which would bias it in favor of F type in males with DUI) or were only gonad samples used? Even if gonad samples were used, depending n their state of maturation/development, they could be heavily dominated by F-type mtDNA.
l. 150. Do not start a sentence with a number. Instead write, “In total, 12 samples…”
l. 156 and following. Does the De novo assembly approach used produce only one dominant assembled genome even if two genomes are present? Have you also considered whether you might be looking at an extremely recently masculinized heteroplasmic male? How would the assembly process treat that case?
l. 295. Write out “coding sequence (CDS)”.
l. 360. Clarify. Do you mean the same number of repeats or the same repeat sequence? CR repeats often evolve by a dynamic process of concerted evolution. Is this likely here too?

As a general comment, this paper makes several fascinating and relevant points regarding the patterns of molecular evolution and codon usage patterns in this genome. It will be interesting for these authors and other DUI investigators to pursue these considerations in future. The authors also make important observations about the level of polymorphism in this species and raise the question as to whether this is a cause or an effect of the ongoing replacement of R. decussatus by the invasive R. phippinarum.

---

## Round 0.2 · accepted · Accept

In this revision, the authors have done a nice job of responding to reviewer's comments. In my opinion, the manuscript is ready for publication. I will send a Word version of the manuscript with some minor editing suggestions which you may choose to incorporate while in production.